# A Mixed-Reality-Based Unknown Space Navigation Method of a Flexible Manipulator

**DOI:** 10.3390/s23083840

**Published:** 2023-04-09

**Authors:** Ronghui Chen, Xiaojun Zhu, Zhang Chen, Yu Tian, Lunfei Liang, Xueqian Wang

**Affiliations:** 1Center of Intelligent Control and Telescience, Tsinghua Shenzhen International Graduate School, Tsinghua University, Shenzhen 518055, Chinawang.xq@sz.tsinghua.edu.cn (X.W.); 2Department of Automation, Tsinghua University, Beijing 100084, China; 3Department of Electronic Engineering, The Chinese University of Hong Kong, Hong Kong, China; 4School of Mechatronics Engineering, Harbin Institute of Technology, Harbin 150001, China

**Keywords:** hyper-redundant manipulator, mixed reality, teleoperation, SLAM, artificial potential field

## Abstract

A hyper-redundant flexible manipulator is characterized by high degree(s) of freedom (DoF), flexibility, and environmental adaptability. It has been used for missions in complex and unknown spaces, such as debris rescue and pipeline inspection, where the manipulator is not intelligent enough to face complex situations. Therefore, human intervention is required to assist in decision-making and control. In this paper, we designed an interactive navigation method based on mixed reality (MR) of a hyper-redundant flexible manipulator in an unknown space. A novel teleoperation system frame is put forward. An MR-based interface was developed to provide a virtual model of the remote workspace and virtual interactive interface, allowing the operator to observe the real-time situation from a third perspective and issue commands to the manipulator. As for environmental modeling, a simultaneous localization and mapping (SLAM) algorithm based on an RGB-D camera is applied. Additionally, a path-finding and obstacle avoidance method based on artificial potential field (APF) is introduced to ensure that the manipulator can move automatically under the artificial command in the remote space without collision. The results of the simulations and experiments validate that the system exhibits good real-time performance, accuracy, security, and user-friendliness.

## 1. Introduction

Nowadays, robotics meets the needs of the manufacturing industry [1], and its use has also been extended to a wider range of fields including medicine, aerospace, and underwater navigation, among others, assisting or replacing human beings in completing some repetitive, heavy, or dangerous tasks. Among them, tasks in complex and unknown spaces, such as earthquake debris rescue, engine room drilling operations, nuclear power plant pipeline maintenance, explosive ordnance disposal, medical procedures (such as endoscopy), and narrow tube welding, continue to pose significant challenges. The complex and unknown spaces share the following characteristics: (i) unstructured environments, (ii) a limited field of view, (iii) unknown internal information, and (iv) relative isolation or danger.

Compared with traditional manipulators with limited degree(s) of freedom (DoF), a hyper-redundant flexible manipulator is distinguished by its better bending properties and flexibility. Specifically, for a rope-driven hyper-redundant flexible manipulator, the motors are uniformly distributed on the base, and the drive ropes drive the rotation of the joints. Thus, this kind of manipulator has a lighter body and can perform well in complex and unknown spaces. However, the ultra-high DoF results in weak spatial perception capability and increases the difficulty of perceiving and judging the environment autonomously and accurately. Therefore, human intervention by means of teleoperation is necessary.

Teleoperation usually employs a bilateral communication method to establish a master-slave control mode, where the operator serves as the master and the remote manipulator as the slave. In this bilateral communication, the master sends commands to the slave while the slave sends feedback back to the master. Traditional teleoperation typically involves the use of computers, joysticks, or other devices to collect operation information, resulting in indirect interaction and low efficiency. Moreover, the types of feedback data are diversified. In addition to the basic states of the manipulator, there is visual feedback, force feedback, etc. [2]. Traditional visual feedback is typically formed by different kinds of cameras and 2D displays, providing the operator with planar visual information. This technology has been developed for more than half a century and has been applied in the fields of hazardous material handling, remote surgery, underwater robots, space robots, and mobile robots [3]. However, traditional 2D visual feedback has some drawbacks, such as a limited field of view, large errors, image degradation, and the inconvenience of attention switching with multi-camera systems [4].

Therefore, the academic community has proposed the concept of immersive interaction, which allows the operator to interact and make decisions in a remote live simulation environment [5]. Visual servoing based on virtual reality (VR), augmented reality (AR), and mixed reality (MR) combined with a head-mounted display (HMD) has good application and development prospects. It can effectively alleviate the degradation of spatial perception caused by the high complexity of the hyper-redundant flexible manipulator [6] and provide an intuitive 3D operating environment for the operator.

This paper presents an MR-based interactive navigation method for hyper-redundant flexible manipulators in an unknown space and establishes a system. The scheme for teleoperation is shown in Figure 1. The contributions of this article are as follows:

First, a novel MR-based teleoperation system frame is put forward. An MR-based interface was developed to offer a virtual model of the remote workspace and virtual interactive interface, allowing the operator to observe the real-time situation from a third perspective and issue commands to the manipulator. To mimic the remote environment, a simultaneous localization and mapping (SLAM) algorithm based on an RGB-D camera is applied. Moreover, a path-finding and obstacle avoidance method based on the artificial potential field (APF) is introduced to ensure that the manipulator can move automatically under the artificial command in the remote space without collision. Additionally, experiments showed that the system met the requirements of real-time performance, accuracy, and user-friendliness.

This paper is arranged as follows: Section 2 summarizes the related work; Section 3 introduces the system design; Section 4 explains the system implementation method, including SLAM and 3D reconstruction, motion control of the manipulator, and the MR-based interface design; Section 5 presents the simulation and experiments carried out; and Section 6 summarizes the work and presents the conclusions.

## 2. Related Work

With high flexibility and environmental adaptability, the hyper-redundant flexible manipulator is often used in environments where operators are difficult to access. Teleoperation technology makes it possible to perform operations remotely and explore places beyond the reach of human power. Nowadays, the teleoperation technology of hyper-redundant flexible manipulators is widely used in the nuclear industry [7,8,9], in underwater operations [10,11], medical surgeries [12,13], the aerospace field [14], and other fields.

Due to diminished environmental awareness, one of the main challenges in teleoperation is to obtain accurate and reliable environmental information feedback. Research experiments have shown that 3D interactive interfaces provide operators with a stronger sense of immersion than 2D interfaces, and effectively help operators understand the environment and make decisions [15]. Therefore, researchers have carried out some immersive teleoperation studies based on VR, AR, and MR. Among them, MR technology pays more attention to the combination of virtual models and real environments. Da Sun et al. [16] proposed a novel multi-DoF manipulator teleoperation system based on MR. By designing the interactive agent of the MR interface, the burden of the operator and the influence of human motion error on the control are reduced. The fuzzy logic control algorithm is applied to adjust the robot’s position and posture, handle potential operational failures, etc. Andrace, Amon [17] proposed an MR-based teleoperation system capable of controlling inspection robots to move heavy objects accurately and robustly and demonstrated the feasibility of MR in performing tasks in remote complex environments.

However, the immersive teleoperation system of hyper-redundant flexible manipulators basically works in known environments. For systems designed to work in unknown spaces, 3D visual SLAM technology has the potential to accomplish remote environmental modeling with semantic information. RGB-D SLAM meets the requirement. Classic RGB-D SLAM schemes include KinectFusion [18], ElasticFusion [19], Kintinuous [20], RGBD SLAM 2 [21], RTAB-Map [22], etc. Among them, the RTAB-Map calculation occupies less memory and has good real-time performance. It uses an efficient memory management mechanism, which greatly reduces the number of nodes required for calculation, and ensures the real-time performance and the accuracy of closed-loop detection. Therefore, it can be applied to 3D mapping of large scenes.

With the MR-based guidance of the operator and real-time environmental reconstruction, the motion of the manipulator can be simplified as a multi-period path-finding and obstacle avoidance problem in a known environmental model. In 1986, Khatib [23] first proposed the APF method for path-finding and obstacle avoidance. The traditional APF method transforms the manipulator into the solution of particles. It is easy to become trapped in the local minimum position, encounter geometric force contradictions, and encounter other related problems. Therefore, it does not apply to hyper-redundant manipulators with a high DoF. The APF method has been optimized for high-DoF manipulators in the academic community. The inverse kinematics was combined with the APF method, the hyper-redundant flexible manipulator was treated as an elastic structure carrying charge, and the obstacles were point charges or line charges [24]. The problem was then transformed into making the key points reach equilibrium in APF. Another solution transformed the obstacle avoidance problem of redundant manipulators into a collision detection problem, focusing on the repulsive field between the obstacle and the two nearest points on the joint [25]. They used Jacobian transpose to construct a virtual relationship between joint torques and then converted the torques into velocities to avoid obstacles. They also introduced the Cartesian space boundary component to optimize the attractive field function, so that the manipulator could reach the target pose at a reasonable speed [26]. Y. Tian et al. [27] proposed an overall planning method for flexible manipulators based on artificial virtual guiding pipelines (VGP) and improved the APF method. The attractive field of VGP was effective in avoiding the problem of being trapped in a local minimum position. Furthermore, the strategy of forcing the manipulator to move in the direction of maximum work within a unit planning period helped avoid complex inverse kinematics, which reduced computational complexity.

## 3. System Design

### 3.1. System Architecture

The system proposed is shown in Figure 2; it was constructed by the MR interactive device, the PC-side control platform, and the remote working space. The operator interacts with the MR interactive device to acquire model information and issue commands through gestures and voice commands. Real-time information transmission is carried out between the MR interactive device and the PC-side control platform. The PC-side control platform processes the environmental and mechanical data collected by the remote flexible manipulator, and sends instructions to the manipulator with the calculation results obtained from built-in algorithms. The manipulator then interacts with the environment to accomplish various tasks.

Microsoft HoloLens II was employed as the MR interactive device. It can scan the operator’s environment in real-time, capture the operator’s displacement, as well as hand and eye movements, and provide voice command services. HoloLens is highly sensitive and operable. In this work, HoloLens displays the virtual model rendering on the PC-side control platform to the operator in the actual scale. The operator can sense and give instructions conveniently with immersive perception.

As the advanced control center of the system, the PC-side control platform is established on Ubuntu 20.04 OS and consists of two components.

The interactive and model control platform is based on Unity 3D, a software tool for 3D model rendering and MR-based application development. The mixed reality toolkit (MRTK) is used to reconstruct the remote workspace and create the MR-based interface. A stable connection between the platform and HoloLens is established via TCP/IP. Additionally, the platform has a motion control module that performs kinematic algorithms, path-finding, and obstacle avoidance algorithms to obtain new joint angles that are within safe limits. The updated calculation result is then transmitted to the virtual model and the manipulator control module.The data processing and control platform is based on the robot operating system (ROS). Unity 3D cannot control the remote hardware directly; therefore, ROS is applied. ROS organizes multiple modules. The RGB-D camera module is responsible for collecting environmental data; the SLAM module performs 3D model reconstruction based on the data collected; the manipulator control module is used for remote hardware control. The information exchange between Unity 3D and ROS is based on the WebSocket protocol.

The flexible manipulator used in this work is depicted in Figure 3. It is a 4 × 4 hyper-redundant flexible manipulator with a total length of 1.52 m and has 16 universal joints that are equally divided into 4 groups. The lengths of small sections in each group of the manipulator are 125 mm, 105 mm, 85 mm, and 65 mm, respectively, and the diameter is 40 mm. Each universal joint in the same group has the same pitch and yaw angles. Therefore, it can be considered that this manipulator with 32 degrees of freedom has 8 effective degrees of freedom. Thus, the flexible manipulator has good flexibility and environmental adaptability when working in a complex space. It can be controlled by the manipulator control module by inputting 8-angle data. An Intel RealSense Depth Camera D435 is mounted on the end effector as the RGB-D camera.

### 3.2. MR-Based Interface

In this work, it is necessary to establish a user-friendly MR-based interface, so that the operator can issue commands more conveniently and efficiently. It is designed in Unity 3D and presented to the operator in HoloLens. The interactive interface consists of three components: an interaction panel, a free target handle, and an anti-shake target handle.

The interaction panel in Figure 4a consists of a main panel and a mode sub-panel; the former contains mode choices and a reset button. Once a mode is chosen, the corresponding sub-panel pops up, which contains diverse control conditions and variables in this mode, providing operators with multiple choices. When the reset button is pressed, the manipulator returns to the initial state. In this work, Mode 3, SLAM Mode, is the navigation system, and its sub-panel includes two control variables: Anti-Shake Handle and Base Lock. The Anti-Shake Handle helps to switch the target handle and the Base Lock determines whether the base is able to move.

Initially, a free target handle is set on the end of the manipulator as shown in Figure 4b, and the operator can drag this handle to adjust the target position and pose for the manipulator to reach. If the Anti-Shake Handle is selected on the sub-panel, the free target handle changes to the anti-shake target handle shown in Figure 4c. The anti-shake target handle is made up of three handles, each extending out of the *x*, *y*, and *z* axes of the target coordinate system. When the operator drags one of the handles, the target end moves in a straight line along the axis, eliminating the offset in other directions caused by handshaking.

Affected by the mechanical characteristics and the step size of the configuration planning, the manipulator’s moving speed is limited. If the target point is too far from the end of the manipulator, the configuration planning algorithm accuracy is reduced, introducing a larger control error. Thus, the system sets an upper limit of $v_{max}$ for the velocity and $s_{max}$ for the distance from the end effector, restricting the operator from dragging the handle in small increments.

## 4. Method

### 4.1. SLAM and 3D Reconstruction

#### 4.1.1. SLAM

To solve the problem of weak environmental awareness of the flexible manipulator, and take advantage of the MR-based immersive operating environment, the system uses the RGB-D camera mounted at the end of the manipulator to scan the environment and performs 3D visual SLAM on the PC-side control platform.

RTAB-Map [22] is used as the SLAM algorithm, which was originally an appearance-based closed-loop detection method and has now evolved into a graph-based SLAM algorithm, with remarkable performance due to its special memory management. The size of the map is always limited, and closed-loop detection can be repeatedly performed within a fixed time limit. Thus, this algorithm can accomplish the goal of long-term and large-scale online scene modeling.

The steps of the RTAB-Map algorithm are as follows: (i) create a location $L_{t}$ using the signature $z_{t}$ of an image and the time index *t*; (ii) update the weight of the acquired location $L_{t}$ and redirect $L_{t}$; (iii) estimate the probability of the hypothesis that $L_{t}$ forms a closed loop based on the discrete Bayesian filter; (iv) select a new closed-loop hypothesis; (v) transfer the neighbors in long-term memory (LTM) of the accepted $L_{t}$ to working memory (WM) after a closed loop is accepted; (vi) transfer the oldest locations with the lowest weight in WM to LTM when the processing time is greater than the threshold.

In this system, the inputs of the SLAM module include RGB-D images collected by the camera, 6 DoF odometry of the camera, and TF representing the relative position of the camera to the base. This input is synchronized with the RTAB-Map SLAM algorithm module to acquire the following outputs: map data with the latest updated node, TF with odometry correction, and a 3D dense point cloud. The SLAM module cooperated well with the RGB-D camera module, and the real-time SLAM result could be monitored in RTAB-Map-viz window.

#### 4.1.2. 3D Reconstruction of the Environment

The reconstruction of the environment includes the reconstruction of two components: camera track and environmentally dense point cloud.

Unity 3D subscribes to the /rtabmap/mapPath and /rtabmap/cloud_map topics published by /rtabmap node in ROS via WebSocket protocol. The former contains a series of camera poses at each sampling point obtained by the SLAM algorithm, and the latter contains the current dense point cloud data generated, including RGB and 3D position coordinates data.

Unity 3D processes the data into Unity 3D coordinates through data type and coordinate system conversion. The camera poses are used for subsequent calibration of joint positioning and obstacle avoidance of the flexible manipulator; the point cloud data are drawn into a colored point cloud in Unity 3D through the built-in mesh tool to reconstruct a virtual 3D model of the remote environment.

### 4.2. Motion Control

According to the mechanical characteristics described in Section 3, the forward and inverse kinematics of the manipulator can be set up. To ensure the efficiency of teleoperation, only an interactive handle at the end of the flexible manipulator is set. The localization of other joints and the overall configuration planning are problems due to the high complexity. Furthermore, with the update of the environment model, the theoretical range of movement is gradually limited, suggesting that the risk of collision with environmental obstacles increases. Therefore, this section establishes an algorithm for autonomous path-finding and obstacle avoidance when the manipulator is guided by the artificial target.

#### Path-Finding and Obstacle Avoidance Method

In the previous work, an overall configuration-planning method based on improved APF [27] is adopted to operate in a known environment. This method introduces an artificial VGP to form an attractive field. Thus, the APF is constructed by three types of potential fields: the attractive potential field pointing to the target position ($field_{att}$), the repulsive potential field away from the obstacles ($field_{rep}$), and the guiding potential field generated by VGP ($field_{Guide}$). *m* key points are sampled on the *n* joint manipulator. The force analysis diagram is presented in Figure 5a. The obstacles are known; $F_{att}$ stands for the force generated by $field_{att}$, $F_{rep}$ presents the force generated by $field_{rep}$, $F_{Guide}$ refers to the force generated by the VGP, $P_{Guide_{i}}\left(i=1,2,\ldots ,N_{Guide}\right)$ refers to the sampling point in VGP where $N_{Guide}$ indicates the number of sampling points in VGP, and $P_{Goal}$ stands for the target point of the end effector.

Nevertheless, this method is only suitable for a known environment where pipeline planning can be done manually in advance. Moreover, the large amount of calculation required can easily lead to system stagnation. Accordingly, the system simplifies and modifies the APF composition to accommodate the shorter planning period. In this condition, the assumption that the environmental model reconstructed by SLAM is a known and stable model in each period of planning is practicable. The simplified APF is constructed by two types of potential fields: $field_{att}$ and $field_{rep}$. The force analysis diagram is presented in Figure 5b. The PCL is the reconstructed virtual environmental model; $F_{att}$ denotes the force generated by $field_{att}$, $F_{rep}$ stands for the force generated by $field_{rep}$, $P_{End}$ stands for the position of the end effector, and $P_{Goal}$ means the target point of the end effector. In order to reach the target pose while reaching the target position, two virtual key points ($P_{End_{1}}$ and $P_{End_{2}})$ are set in the system, respectively, at the unit lengths of the *x* and *y* axes of the tool coordinate system, which are in the attractive potential field generated by two virtual points, $P_{Goal_{1}}$ and $P_{Goal_{2}}$, respectively, at the unit length of the *x* and *y* axes of the target coordinate system. The formula for APF is as follows:(1)$$F_{att}=\left\{\begin{array}{ccc} k_{att}\left(P_{Goal}-P\right), & \; & |P_{Goal}-P|\leq d_{1}\\ \frac{k_{att}\left(P_{Goal}-P\right)}{|P_{Goal}-P|}, & \; & d_{1}<\left|P_{Goal}-P\right|\leq d_{2}\\ 0, & \; & |P_{Goal}-P|>d_{2} \end{array}\right.$$
(2)$$F_{rep}=\left\{\begin{array}{ccc} k_{rep}{\left(\frac{1}{|P-P_{0}|}-\frac{1}{\rho }\right)}^{2}\cdot \frac{P-P_{0}}{|P-P_{0}|}, & \; & |P-P_{0}|\leq \rho \\ 0, & \; & |P-P_{0}|>\rho \end{array}\right.$$
where $k_{att},k_{rep},d_{1},d_{2}$ are positive definite coefficients, *P* stands for a key point on the manipulator in APF, $P_{0}$ stands for the nearest point to *P* in PCL.

To apply the APF construction to the overall configuration planning, the maximum-work planning strategy motivates each key point to move in the direction of the maximum sum of work in the APF in a unit-planning period, thus avoiding using the complex inverse kinematics for a flexible manipulator.

In this configuration planning problem, joint angles $\theta_{i}$ and $\phi_{i}\left(i=1,2,...,n\right)$, the displacement of the base $\left(d_{x},d_{y},d_{z}\right)$ and the rotated Euler angles (α,β,γ) form a (2n+6)-dimensional state variable x. The specific principle is as follows.

For each key point $P_{i}\left(i=1,2,...,m\right)$, the force in a planning period is calculated: (3)$$F_{i}=F_{att_{i}}+F_{rep_{i}}.$$

The sum of work can be expressed as: (4)$$w_{i}=F_{i}\cdot dP_{i}$$
(5)$$W=\sum_{i=1}^{n+2}F_{i}^{T}\cdot dP_{i}.$$

The partial derivative of the total work concerning the state variable is obtained: (6)$$\frac{\partial W}{\partial \vec{x}}=\sum_{i=1}^{n+2}F_{i}^{T}\cdot \frac{\partial P_{i}}{\partial \vec{x}}$$
where $\frac{\partial P_{i}}{\partial \vec{x}}$ is the Jacobian Matrix at $P_{i}$, formulated as $J_{i}$. Then the total work is sorted out: (7)$$W=\sum_{i=1}^{n+2}F_{i}^{T}J_{i}\Delta x.$$

Since the planning is strictly limited to a short planning period, the planning step size needs to be strictly limited to prevent problems, such as collision or uncontrollability caused by large errors. The planning problem is eventually converted to a convex optimization problem, expressed as follows: (8)$$\begin{array}{cc} \; & \max W=\sum_{i=1}^{n+2}F_{i}^{T}J_{i}\Delta x\\ \; & \begin{array}{ccc} s.t. & \;|J_{i}\cdot \Delta x|\leq a, & \left(i=1,2,...,n+2\right)\\ \; & \;|\Delta x_{i}|\leq b_{i}, & \left(i=1,2,...,n+2\right)\\ \; & \;|x_{i}+\Delta x_{i}|\leq c_{i}, & \left(i=1,2,...,n+2\right)\\ \; & \;OtherConstraints \end{array} \end{array}$$
where *a* and $b_{i}$ stand for the upper limit of the displacement of each key point and a single step in a unit planning period, respectively, both of which are positively correlated with the shortest distance to the PCL and the distance to the target point; $c_{i}$ stands for the upper limit of state variable x, which is only relevant to the manipulator; OtherConstraints helps to accomplish more functions, such as locking the movement of the base or the end effector.

## 5. Evaluation Results

To evaluate the system performance, the experiment was divided into three parts: SLAM and 3D reconstruction simulation, configuration planning simulation, and MR-based teleoperation experiment.

### 5.1. SLAM and 3D Reconstruction Simulation

In this part, SLAM and 3D reconstruction simulations were carried out. The cooperation between the RGB-D camera and the PC-side control platform was validated, and the effectiveness of the remote environmental reconstruction was demonstrated. The experimental setup consisted of the RealSense D435 depth camera and a computer with a Ubuntu 20.04 system installed with Unity 3D and ROS.

The SLAM results in the RTAB-Map-viz and the 3D reconstruction in the Unity 3D were recorded when the operator moved around with the camera in hand. The environmental reconstruction process is presented in Figure 6.

Apart from the environmental reconstruction, the camera track was also estimated during the SLAM process. Figure 7 shows the final complete 3D reconstruction results, including the overall environmental model and the camera track in Unity 3D, where the blue point set represents the camera track points at each sampling point, and the colored point set represents the environment model. Moreover, the update period of the point cloud reconstruction was also tested, and the average update period was 1.203 s.

The SLAM and reconstruction process demonstrated good real-time performance and efficient acquisition of environmental information and camera motion data. Moreover, the semantic information obtained can assist the operator in effectively recognizing the environment.

### 5.2. Configuration Planning Simulation

In this simulation, the configuration planning method was tested in Unity 3D based on an already reconstructed environmental model. The operator artificially planned a trajectory of the target end in advance and recorded the movement of the manipulator through the point cloud.

Figure 8a shows that the manipulator could follow the artificial command, eventually reaching the target end, while avoiding the obstacles at the same time. Additionally, when Base Lock was chosen, the movement process with an artificially planned trajectory of the target end is presented in Figure 8b. The average planning cycle was tested to be 93.23 ms, indicating the feasibility and effectiveness of the overall configuration planning method.

Furthermore, this process was tested with HoloLens connected. The average time for rendering a single frame in HoloLens was 65.37 ms, which is 15.30 frames/s. This time is shorter than the average time of the configuration planning cycle, because the configuration planning runs on a separate thread and the movement of the virtual manipulator in each frame is the result of a simple linear interpolation process.

### 5.3. MR-Based Teleoperation Experiment

This experiment aimed to verify that the system could provide an immersive MR operating environment for the operator. This experiment was carried out based on HoloLens and a computer. The operator wore HoloLens and operated in the MR scene.

First, the operator dragged the free target handle to adjust the end effector’s position and pose. The manipulator changed the configuration to follow the command through forward and inverse kinematics. Figure 9 presents the real scene of the MR-based teleoperation.

Second, the operator dragged the free target handle to form a VGP through known obstacles. The configuration planning based on APF and VGP was performed to instruct the manipulator to find the path and avoid obstacles. We set the HoloLens on a fixed observing point and recorded the path-finding process, which is shown in Figure 10. In Figure 10a, the red dashes stand for the 16 DoF flexible manipulator; the black dashes stand for the VGP; the green dot and the two vertical line segments at the end of the VGP stand for the target position and pose of the end; and the remaining 3D shapes are the obstacles in the environment. Furthermore, the whole model is established and transformed into the HoloLens in Figure 10b.

Based on the above experimental results, it was proved that Unity 3D and HoloLens could build a stable information connection. Operators wearing HoloLens can perceive the virtual model of the remote manipulator and its environment, and intuitively issue commands to the manipulator through the interactive interface. Therefore, the effectiveness of the MR-based unknown space navigation system for the flexible manipulator has been verified.

## 6. Conclusions

In this work, a novel MR-based flexible manipulator navigation system for unknown spaces is designed. The operation and information transmission are organized in a reasonable and orderly manner, linking manipulator control with real-time environmental modeling, which assists the operator in perceiving the complex unknown space and issuing commands from a first-person perspective. Moreover, SLAM and configuration planning are adapted in this work. The former efficiently renders the environment model to the operator, and the latter helps the manipulator find paths to an artificial target without collision. Simulations and experiments validated the effectiveness of the system and the user-friendliness of the interaction. It is believed that this work can lay a foundation for future work in MR-based flexible manipulator teleoperation.

However, we have not yet considered the dynamics of the manipulator or fully leveraged the advantages of MR. In the future, we will consider better combining the virtual world and the real world by introducing a manipulator as the master instead of the virtual manipulator in Unity 3D. Moreover, the reconstruction and configuration planning methods deserve further optimization in terms of real-time performance and accuracy. Additionally, in the future, the system can be applied to real manipulators and accomplish teleoperation in space tasks, underwater exploration, and so on.

## Figures and Tables

**Figure 1 sensors-23-03840-f001:**
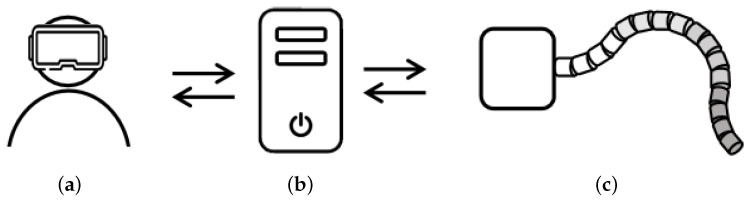
Scheme for teleoperation. (**a**) Operator and the MR interactive device. (**b**) PC-side control platform. (**c**) Flexible manipulator with an RGB-D camera.

**Figure 2 sensors-23-03840-f002:**
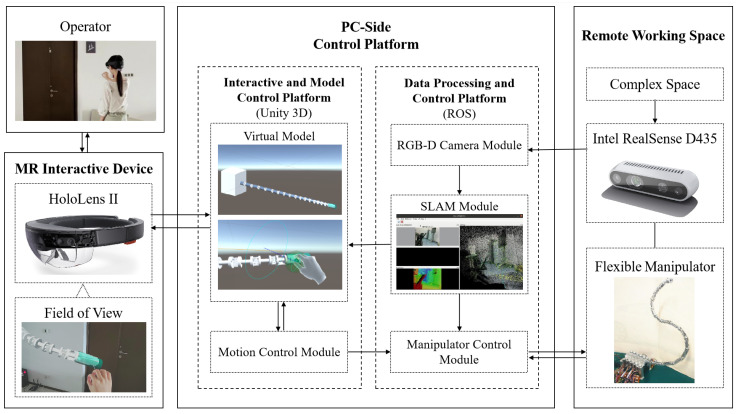
System architecture.

**Figure 3 sensors-23-03840-f003:**
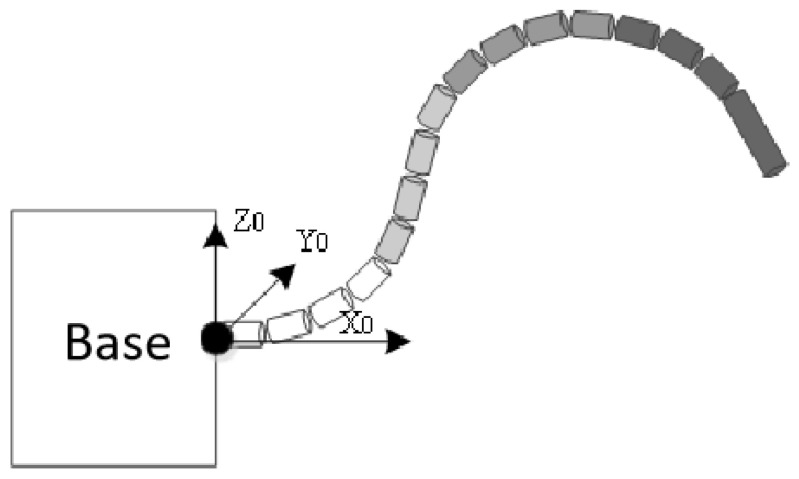
Hyper-redundant flexible manipulator.

**Figure 4 sensors-23-03840-f004:**
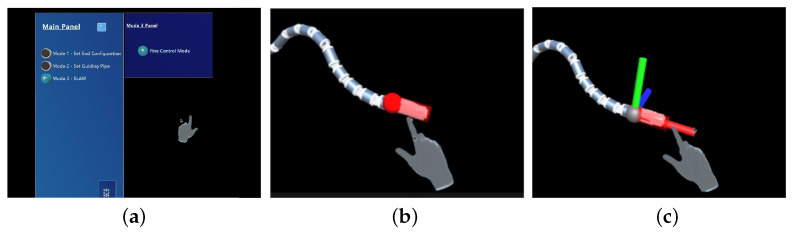
The MR-based interface. (**a**) Interaction panel. (**b**) Free target handle. (**c**) Anti-shake target handle.

**Figure 5 sensors-23-03840-f005:**
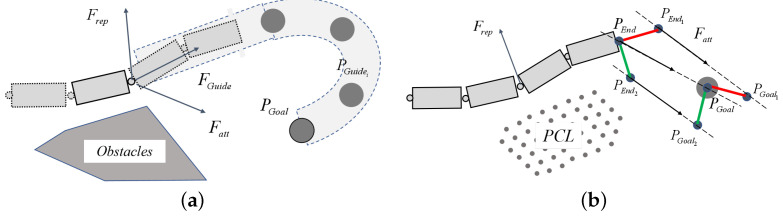
Force analysis diagram in two kinds of APF: (**a**) APF with guiding potential field; (**b**) Simplified and modified APF.

**Figure 6 sensors-23-03840-f006:**
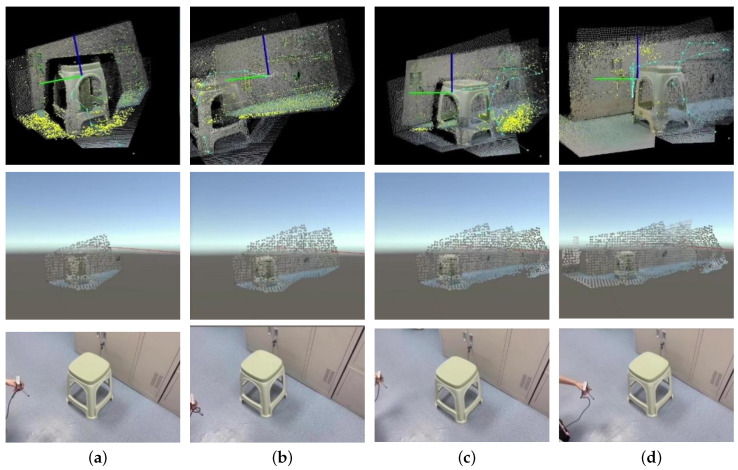
Simulation and experiment results of the SLAM and 3D reconstruction. The first row shows the real-time SLAM results in RTAB-Map-viz. The second row shows the real-time 3D point cloud reconstructed in Unity 3D. The third row shows the pose of the handheld Intel RealSense D435. (**a**) $t=t_{1}$. (**b**) $t=t_{2}$. (**c**) $t=t_{3}$. (**d**) $t=t_{4}$.

**Figure 7 sensors-23-03840-f007:**
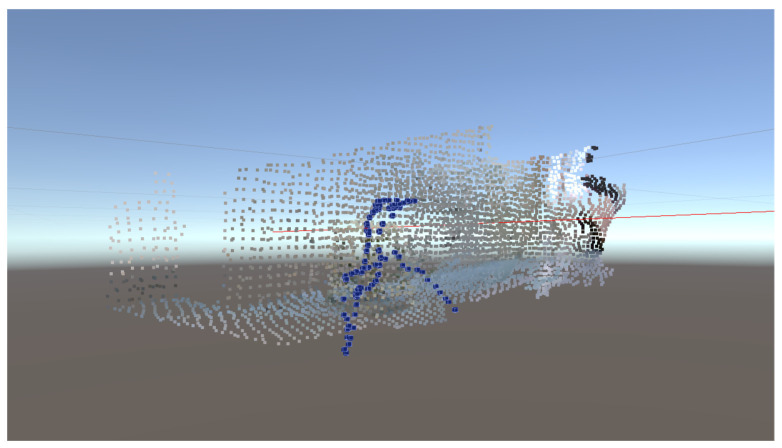
Final complete 3D reconstruction results, including the overall environmental model and the camera track in Unity 3D.

**Figure 8 sensors-23-03840-f008:**
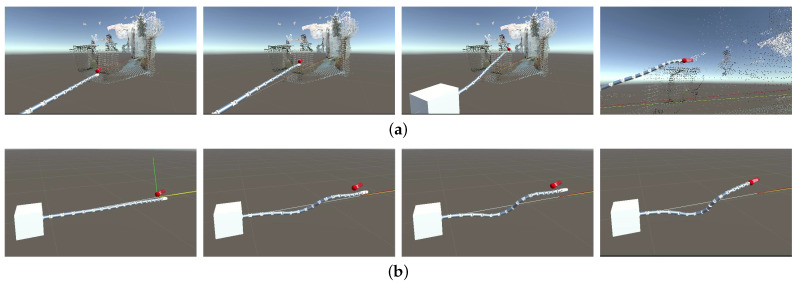
Results in chronological order of the (**a**) configuration planning simulation and (**b**) configuration planning simulation with the Base Lock option chosen.

**Figure 9 sensors-23-03840-f009:**
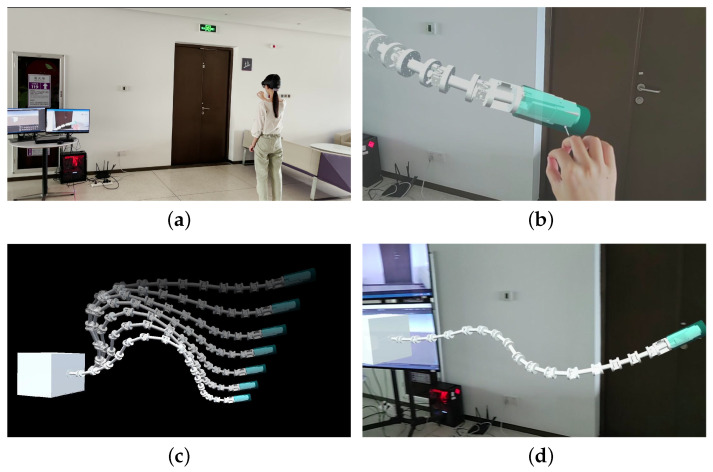
MR-based teleoperation experiment 1. (**a**) Operator in the experimental scene. (**b**) Operator’s field of view. (**c**) Simulation of the manipulator’s kinematics. (**d**) Virtual model observed in HoloLens.

**Figure 10 sensors-23-03840-f010:**
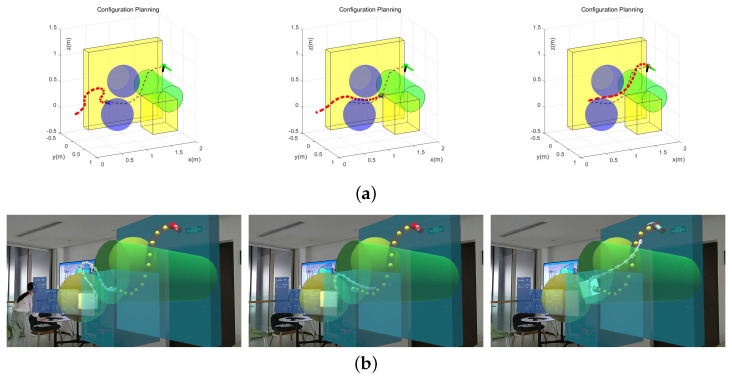
The results in chronological order of the (**a**) configuration planning simulation and (**b**) configuration planning simulation with the Base Lock option chosen.

## Data Availability

Not applicable.

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
