# Peer review of "A Mixed-Reality-Based Unknown Space Navigation Method of a Flexible Manipulator"

_sensors, 2023, doi:10.3390/s23083840_

Round 1
Reviewer 1 Report
The authors designed an interactive navigation method based on mixed reality of a hyper-redundant flexible manipulator in a unknown space. Through a novel teleoperation system, an MR-based interface is developed to provide a virtual model of the remote workspace and virtual interactive interface, allowing the operator to observe the real-time situation from a third perspective and issue commands to the manipulator. The work is very interesting and the paper is well writen, with almost no errors found. The next comments has as objective to improve paper quality and readers' understanding of the ideas presented.
The first point I would like to discuss is about the real necessity of MR and a Mixed-reality device such as Hololens in this project. The authors should discuss about using other devices such as an Oculus Quest 2, since it can also track the user hands for manipulation and counts with a inside-out tracking system capable of capturing the user's 6DoF information.
Its hard to understand Figure 7 as it is much different from the ones from Figure 6. Please explain what the reader should see in Figure 7.
When the authors talk about "real time performance", what does this represent in terms of frames per second? What is the lag between the command issued by the operator and the manipulator to execute the activity?
Please provide more details about the PC-side configuration necessary to run the proposed solution and how fast it run.
"Moreover, the reconstruction and configuration planning methods deserve further optimization in terms of real-time performance and accuracy. " -> what is the current performance and accuracy? This is not detailed in the text.
It would be nice to have more details regarding the tests that validated the proposed system. How many users were involved in the tests? What did they think about the experience?
Is the teleoperation performed over the virtual robot or does it control the real one? Please provide more information regarding the real manipulator and images/videos of the final result of the project.
More general comments and a few minor errors are listed as follows.
There are no a, b and c indications on Figure 1.
"CC." -> ?
"WR." -> ?
"voices command." -> "voice commands."
Reviewer 2 Report
The authors designed an interactive navigation method based on mixed reality (MR) of a hyper-redundant flexible manipulator (snake robot) with high degrees of freedom in an unknown space. MR-based interface is developed to observe the real-time situation from a third perspective. , a path-finding and obstacle avoidance method based on artificial potential field (APF) is applied for autonomous motion.
The overall work is very interesting but lots of literature appears to be shallow. It is recommended to consider some recent literature on “optimized Mobile robot control and navigation in unknow space” and “Challenges and opportunities on AR/VR technologies”
Eswaran, M., et al. " Augmented reality-based guidance in product assembly and maintenance/repair perspective: A state of the art review." Expert Systems with Applications (2022): 118983.
Chang, Lu, et al. "Reinforcement based mobile robot path planning with improved dynamic window approach in unknown environment." Autonomous Robots 45 (2021): 51-76.
Figure 1(c ), the term flexible manipulator and the illustration does not match.
Please provide a separate section for abbreviations
In general, the device Intel Realsence D435 is large; how can it be assembled with the flexible manipulator to explore complex workspace (Like pipe inspection as stated in the introduction)
Section 4.2.1 seems to be modelling for a known environment,
What kind of limitation exists during Realtime implementation, it seems rendering is not clear in figure 8(a), would the user be able to see the scene correctly?
What is the response time of the whole mechanism?
Please expand the conclusions with future scope.
Round 2
Reviewer 1 Report
I'm satisfied with the modifications performed by the authors. Congratulations, I believe the paper is ready for acceptance now.